# Behavioral-state modulation of inhibition is context-dependent and cell type specific in mouse visual cortex

Janelle MP Pakan[1], Scott C Lowe[2], Evelyn Dylda[1], Sander W Keemink[2,3], Stephen P Currie[1], Christopher A Coutts[1], Nathalie L Rochefort[1]*

[1]Centre for Integrative Physiology, School of Biomedical Sciences, University of Edinburgh, Edinburgh, United Kingdom; [2]Institute for Adaptive and Neural Computation, School of Informatics, University of Edinburgh, Edinburgh, United Kingdom; [3]Bernstein Center Freiburg, Faculty of Biology, University of Freiburg, Freiburg, Germany

**Abstract** Cortical responses to sensory stimuli are modulated by behavioral state. In the primary visual cortex (V1), visual responses of pyramidal neurons increase during locomotion. This response gain was suggested to be mediated through inhibitory neurons, resulting in the disinhibition of pyramidal neurons. Using in vivo two-photon calcium imaging in layers 2/3 and 4 in mouse V1, we reveal that locomotion increases the activity of vasoactive intestinal peptide (VIP), somatostatin (SST) and parvalbumin (PV)-positive interneurons during visual stimulation, challenging the disinhibition model. In darkness, while most VIP and PV neurons remained locomotion responsive, SST and excitatory neurons were largely non-responsive. Context-dependent locomotion responses were found in each cell type, with the highest proportion among SST neurons. These findings establish that modulation of neuronal activity by locomotion is context-dependent and contest the generality of a disinhibitory circuit for gain control of sensory responses by behavioral state.

**\*For correspondence:**
n.rochefort@ed.ac.uk

**Competing interests:** The authors declare that no competing interests exist.

## Introduction

Sensory perceptions are modulated by the context in which they are experienced. In primary sensory areas, neuronal responses to sensory inputs are also modulated by behavioral states, including level of arousal, attention and locomotion (*Iriki et al., 1996*; *Petersen and Crochet, 2013*; *Bennett et al., 2014*; *McGinley et al., 2015*). In vivo recordings in awake mice have shown that locomotion modulates the response properties of neurons in the primary visual cortex (V1), resulting in an increased gain of excitatory neuron responses to visual stimuli (*Niell and Stryker, 2010*; *Keller et al., 2012*; *Bennett et al., 2013*; *Polack et al., 2013*; *Saleem et al., 2013*; *Erisken et al., 2014*; *Reimer et al., 2014*). However, the neuronal circuits underlying this response modulation are unclear.

Recent studies have revealed that a specific subclass of inhibitory neurons, expressing vasoactive intestinal peptide (VIP), strongly increase their activity during locomotion (*Fu et al., 2014*; *Reimer et al., 2014*; *Jackson et al., 2016*). VIP neurons mainly inhibit a second class of inhibitory neurons, expressing somatostatin (SST; *Figure 1A*; *Pfeffer et al., 2013*; *Jiang et al., 2015*; *Urban-Ciecko and Barth, 2016*). It has been proposed that cholinergic activation of VIP neurons during locomotion would inhibit SST neurons, alleviating inhibition onto excitatory neurons and, as a consequence, increase the gain of excitatory neuron visual responses (*Figure 1B*; *Fu et al., 2014*). However, a previous study has reported an increase of SST spiking activity in layer 2/3 during locomotion (*Polack et al., 2013*), an observation that challenges the hypothesis of an SST-cell mediated

**eLife digest** How we perceive what we see depends on the context in which we see it, such as what we are doing at the time. For example, we perceive a park landscape differently when we are running through it than when we are sitting on a park bench. Behavior can also alter neuronal responses in the brain. Indeed, the neurons in the part of the brain that receives information related to vision (known as the visual cortex) respond differently to visual stimuli when an animal is moving compared to when the animal is still. However, while some recent studies revealed that specific types of neurons become more or less responsive during movement, others reported the opposite results.

One hypothesis that would explain these contradictory findings would be if the way that behavior, in this case movement, affects neuronal responses also depends on the external context in which the movement happens. Now, Pakan et al. have tested this hypothesis by imaging the activity of different types of neurons in the primary visual cortex of mice that were either running on a treadmill or staying still. The experiments were conducted in two different contexts: in total darkness (in which the mice could not see) and in the presence of display screens (which provided the mice with visual stimulation).

Pakan et al. confirmed that running does indeed affect the activity of specific neurons in different ways in different contexts. For example, when the mice received visual stimulation, the three main classes of neurons that send inhibitory signals in the visual cortex became more active during running. However, when the mouse ran in the dark, two of these neuron types became more active during running while the third type of neuron was unresponsive. This finding reveals more about the dynamic nature of inhibitory activity that strongly depends on the animal's behaviour. It also shows how these neurons influence the excitatory neurons in the visual cortex, which send information to the rest of the brain for further processing towards perception.

The next step will be to identify what precise mechanism makes these neurons respond differently in unique contexts, and to tease apart how these movement-dependent signals affect the way animals perceive visual stimuli.

disinhibitory circuit. The aforementioned recordings of SST neuronal activity were acquired in different sensory contexts, either in darkness or during the presentation of visual stimuli. One hypothesis that would explain the discrepancies between these results is that V1 neuronal responses to locomotion are context-dependent.

In this study, we tested this hypothesis by directly comparing the locomotion responses of excitatory and inhibitory neurons in darkness and during visual stimulation. We used two-photon calcium imaging to monitor the activity of excitatory neurons as well as of three non-overlapping populations of inhibitory neurons (VIP, SST and parvalbumin [PV] neurons) in layer 2/3 and layer 4 of V1 in awake behaving mice. Our results show that during visual stimulation these three classes of interneurons increase their activity with locomotion, challenging the model of a disinhibitory circuit mediated through SST neurons. We found that locomotion affects the activity of inhibitory circuits differently in darkness and during visual stimulation, revealing a context-dependent, cell type specific response to locomotion in V1. The highest proportion of context-dependent responses to locomotion was found among SST neurons, which play a central role in V1 microcircuits. We suggest alternative mechanisms of how locomotion modulates the neuronal activity in V1, highlighting the dynamic nature of interneurons function that strongly depends on the behavioral context of the animal.

## Results

We compared the modulation of neuronal activity by locomotion in the mouse primary visual cortex (V1), between two different sensory contexts: darkness and visual stimulation. To do this, we used two-photon calcium imaging in head-fixed mice that ran freely on a cylindrical treadmill (*Figure 1C*). The relative changes in somatic fluorescence of the genetically-encoded calcium indicator GCaMP6f were used as a non-linear readout of the neuronal spiking activity (*Chen et al., 2013*). Inhibitory neuronal subtypes were labeled by injecting adeno-associated viruses (AAVs) into V1 of Cre-

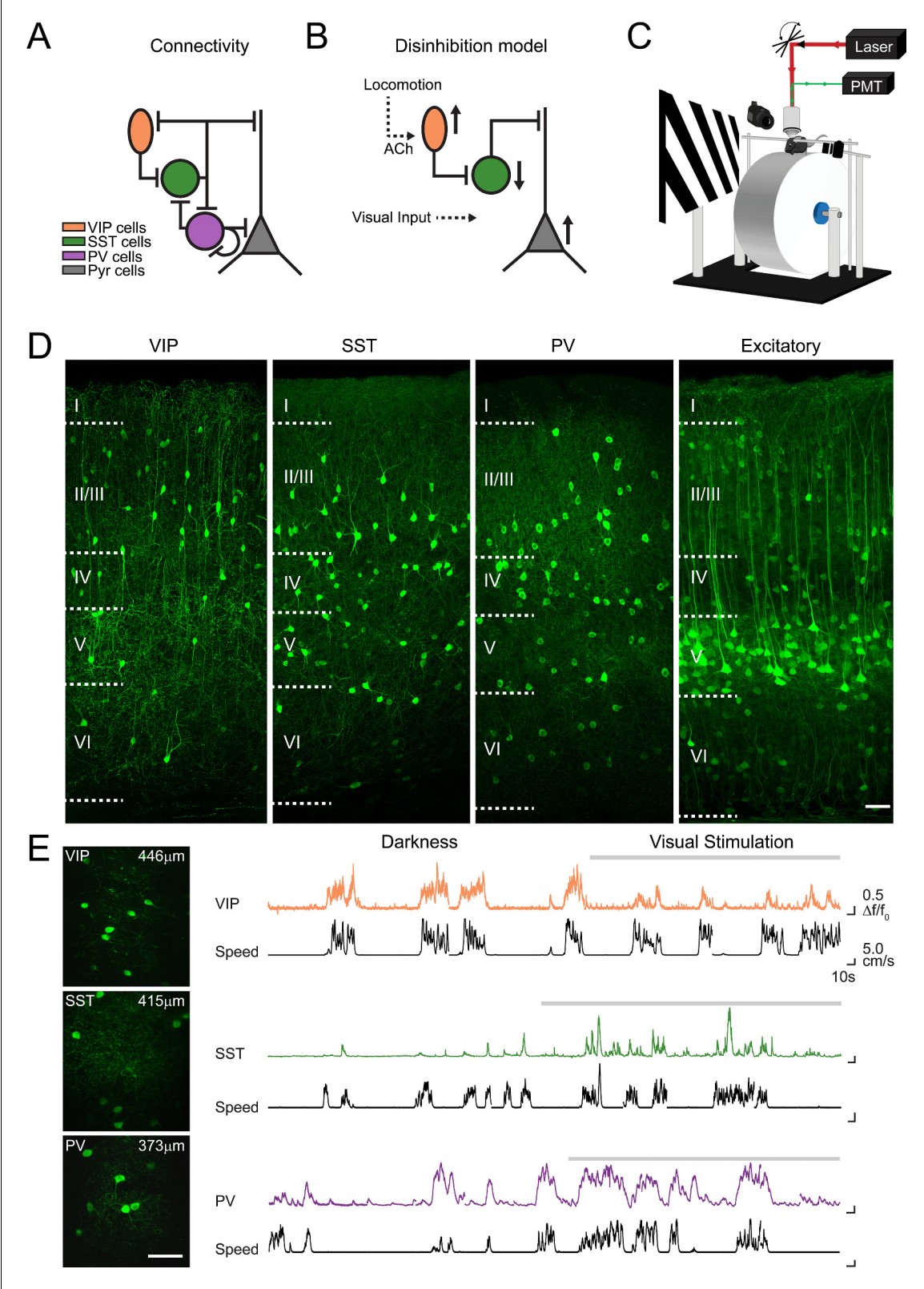

**Figure 1.** Imaging locomotion responses of excitatory and inhibitory neurons in mouse V1. (**A**) Schematic of the connectivity between pyramidal neurons (Pyr) and subtypes of inhibitory neurons, vasoactive intestinal peptide (VIP), somatostatin (SST) and parvalbumin (PV) expressing neurons, established from in vitro studies in V1 (*Pfeffer et al., 2013*; *Jiang et al., 2015*). (**B**) Proposed disinhibition model: locomotion activates VIP neurons through cholinergic (ACh) inputs, SST neurons are inhibited, which leads to a disinhibition of Pyr neurons and an increase in the gain of visual responses

*Figure 1 continued on next page*

*Figure 1 continued*

during locomotion (*Fu et al., 2014*). (C) Experimental set-up for two-photon calcium imaging in V1 of awake-behaving mice. Mice are head-fixed and can run freely on a cylindrical treadmill either during the presentation of a visual stimulus (oriented gratings) or in darkness. (D) Confocal images of 50 μm thick coronal sections showing cell type specific GCaMP6f expression in VIP, SST and PV-positive inhibitory neurons as well as in CaMKII-positive excitatory populations. Boundaries between cortical layers are indicated. (E) Left panel, in vivo two-photon images of VIP, SST and PV neurons labelled with GCaMP6f; cortical depth of imaging is indicated. Right panel, example calcium transients (ΔF/F0, coloed traces) of single VIP, SST and PV neurons, imaged in darkness and during visual stimulation with oriented gratings (grey bar above trace), and aligned with the corresponding running speed (cm/s, black traces). Scale bars on images, 50 μm.

recombinase transgenic mice (PV-, SST-, or VIP-Cre mice) for the Cre-inducible expression of the genetically-encoded calcium indicator GCaMP6f (*Figure 1D–E*; *Chen et al., 2013*). To image excitatory neurons, we co-injected a floxed version of GCaMP6f and an AAV where Cre expression is driven by a CaMKII promoter, into C57/BL6 mice. After 2–3 weeks of expression, we recorded the running speed and GCaMP6f signals simultaneously, both in total darkness and during visual stimulation (*Figure 1E*).

## Layer 2/3 celltype-specific responses to locomotion differ in darkness and during visual stimulation

### Excitatory neurons

We quantified, for each excitatory neuron (n = 1124 in 12 mice), the mean amplitude of calcium transients during locomotion periods and stationary periods, both during visual stimulation (drifting gratings) and in darkness (*Figure 2A(i),B(i)*). In agreement with previous electrophysiological observations (*Niell and Stryker, 2010*; *Keller et al., 2012*; *Bennett et al., 2013*; *Polack et al., 2013*; *Saleem et al., 2013*; *Erisken et al., 2014*; *Reimer et al., 2014*), we observed that, on average, locomotion increased the amplitude of calcium transients in excitatory neurons during visual stimulation (*Figure 2B(i)*, *Figure 2—figure supplement 1B(i)* mean change in fluorescence [$\Delta F/F_0$] = 0.12 ± 0.02 locomotion versus 0.07 ± 0.01 stationary; p<0.001, n = 12, Wilcoxon signed rank test). We quantified the effect of locomotion by calculating a locomotion modulation index (LMI) for each neuron, corresponding to the difference between the mean $\Delta F/F_0$ during locomotion ($R_L$) and stationary ($R_s$) periods, normalized by the sum of the mean $\Delta F/F_0$ during both behavioral states (LMI = $(R_L - R_s)/(R_L + R_s)$). An LMI equal to 0 indicates no difference between locomotion and stationary periods, while an LMI equal to 0.5 indicates that the average amplitude of calcium transients was three times higher during locomotion than during stationary periods. Comparing the distribution of LMIs between the two sensory contexts, we found that the modulation of the activity of excitatory neurons by locomotion was significantly different in darkness compared to visual stimulation (*Figure 2C(i),D(i)*; mean of median LMI: 0.07 ± 0.02 darkness versus 0.19 ± 0.02 visual stimulation; p=0.001, n = 12, Kruskal–Wallis test). During visual stimulation, 47 ± 4% of excitatory neurons were significantly locomotion responsive (see Materials and methods for locomotion responsive criteria), compared with 28 ± 4% in darkness. Additionally, in the dark, a small proportion of neurons were inhibited by locomotion, decreasing their activity during locomotion periods relative to stationary periods (10 ± 1% of neurons).

### VIP neurons

As reported in previous studies (*Fu et al., 2014*; *Reimer et al., 2014*; *Jackson et al., 2016*), we found that VIP neurons (n = 210 in 12 mice) strongly responded to locomotion (*Figure 1E* and *Figure 2A(ii),B(ii)*). This was true both in darkness (mean $\Delta F/F_0$ = 0.51 ± 0.12 locomotion versus 0.10 ± 0.03 stationary; p<0.001, n = 12, Wilcoxon signed rank test) as well as during visual stimulation (mean $\Delta F/F_0$ = 0.42 ± 0.14 locomotion versus 0.09 ± 0.02 stationary; p<0.001, n = 12) with no significant difference in the average LMI between sensory contexts (*Figure 2C(ii),D(ii)*, mean of median LMI: 0.60 ± 0.05 darkness versus 0.49 ± 0.06 visual stimulation; p=0.106, n = 12, Kruskal–Wallis test; see also *Figure 2—figure supplement 1B(ii)*). A high proportion of VIP neurons were significantly locomotion responsive in both sensory contexts (85 ± 7% in darkness and 79 ± 6% during visual stimulation).

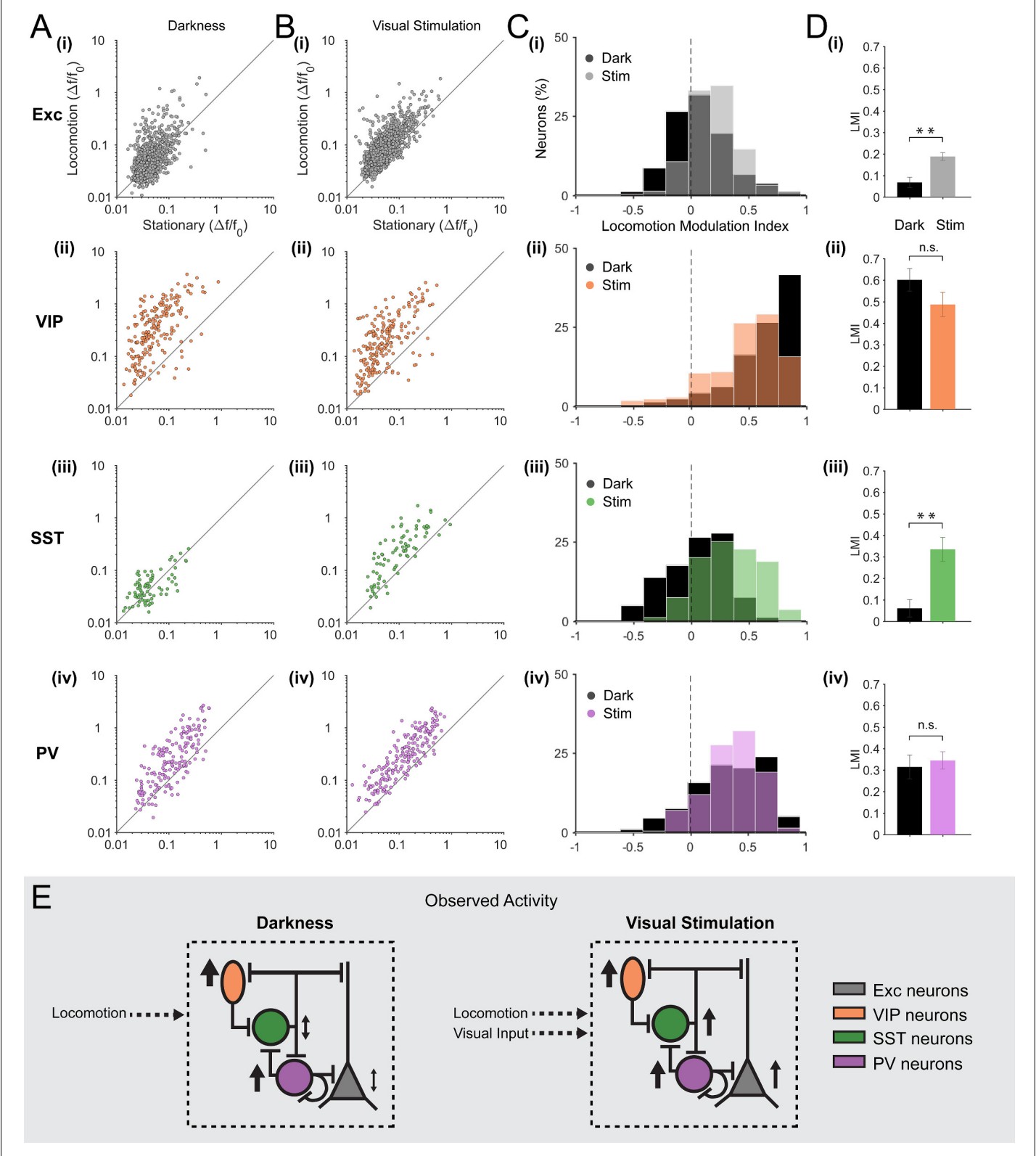

**Figure 2.** Locomotion differentially modulates excitatory and inhibitory neuronal responses in darkness and during visual stimulation in V1 layer 2/3. (A–B) Scatter plots of the mean amplitude of fluorescence changes ($\Delta F/F_0$) of each neuron for locomotion periods versus stationary periods, in darkness (A) and during visual stimulation (oriented gratings) (B); (i) excitatory cells (Exc), n = 1124; (ii) VIP, n = 210; (iii) SST, n = 79; (iv) PV, n = 199 neurons. (C) Histograms of the distribution of locomotion modulation indices (LMI = ($R_L$ – $R_S$)/($R_L$ + $R_S$), where $R_L$ and $R_S$ are the mean $\Delta F/F0$ during locomotion and

*Figure 2 continued on next page*

*Figure 2 continued*

stationary periods, respectively), for each cell type, in darkness (Dark, black) and during visual stimulation (Stim, coloed). An LMI equal to 0 indicates no difference between locomotion and stationary periods, while an LMI equal to 0.5 indicates that the average amplitude of calcium transients was three times higher during locomotion than during stationary periods. (D) Mean of the median LMI per animal and s.e.m. **p<0.01, n.s., not significant (p>0.05); n = 12 (i), 12 (ii), 11 (iii), 13 (iv) mice; Kruskal–Wallis test. (E) Schematic representation of the results. Size and direction of the arrows indicate the average response per cell type during locomotion (increasing or decreasing activity). In darkness, SST and excitatory neurons were largely non-responsive to locomotion while VIP and PV neurons were strongly activated by locomotion. However, during visual stimulation, locomotion increases the responses of excitatory neurons as well as of the three classes of inhibitory neurons (VIP, SST and PV).

The following figure supplements are available for figure 2:

**Figure supplement 1.** Modulation of excitatory and inhibitory neurons responses by locomotion during the presentation of patterned (oriented gratings) and non-patterned (grey screen) visual stimuli.

**Figure supplement 2.** Visual responsiveness of excitatory and inhibitory neurons during stationary and locomotion periods.

**Figure supplement 3.** Cross correlation of fluorescence changes ($\Delta F/F_0$) with running speed.

In order to compare our results directly with previous findings (*Fu et al., 2014*), we calculated the cross-correlation between VIP calcium signals and running speed. We confirmed the presence of a single positive peak around time zero, both in darkness and during visual stimulation (*Figure 2—figure supplement 2A(ii)*). We also observed a lower amplitude during visual stimulation but this decrease was not significant (mean zero-time correlation: 0.26 ± 0.04 in darkness versus 0.20 ± 0.02 during visual stimulation; p=0.225, n = 12, Kruskal–Wallis test; *Figure 2—figure supplement 2C*). Similarly, the mean $\Delta F/F_0$ (*Figure 2—figure supplement 3C(ii)*) and the mean LMI (*Figure 2D(ii)*) of VIP neurons also decreased during visual stimulation, without reaching significance (mean $\Delta F/F_0$ = 0.51 ± 0.12 in darkness versus 0.42 ± 0.14 during visual stimulation; p=0.151, n = 12, Wilcoxon signed rank test).

## SST neurons

In contrast to VIP neurons, responses of SST neurons (n = 79 in 11 mice) to locomotion were found to be highly context-dependent. During visual stimulation, the mean $\Delta F/F_0$ during locomotion periods was significantly higher than during stationary periods (*Figure 2B(iii)*, *Figure 2—figure supplement 1B(iii)*; mean $\Delta F/F_0$ = 0.25 ± 0.05 locomotion versus 0.10 ± 0.03 stationary; p=0.001, n = 11, Wilcoxon signed rank test). However, in darkness, SST neurons were either non-responsive, increased or even decreased their activity during locomotion with, on average, no significant difference between locomotion and stationary periods (*Figure 2A(iii)*, *Figure 2—figure supplement 1B (iii)*; mean $\Delta F/F_0$ = 0.06 ± 0.02 locomotion versus 0.06 ± 0.01 stationary; p=0.102, n = 11, Wilcoxon signed rank test). As a result, the modulation of SST neuron responses by locomotion was found to be significantly different across sensory contexts (*Figure 2C(iii),D(iii)*, mean of median LMI: 0.06 ± 0.04 darkness versus 0.33 ± 0.06 visual stimulation; p=0.002, n = 11, Kruskal–Wallis test). During visual stimulation, 63 ± 7% of SST neurons were significantly locomotion responsive (increasing their activity) and only 4 ± 3% were decreasing their activity during locomotion. In darkness, the percentage of neurons increasing their activity dropped to 24 ± 6% with an additional 11 ± 5% of SST neurons decreasing their activity during locomotion.

In line with these results, the cross-correlation between SST calcium transients and running speed significantly increased during visual stimulation compared to darkness (mean zero-time correlation = 0.04 ± 0.01 in darkness versus 0.13 ± 0.01 during visual stimulation; p=0.001, n = 11, Kruskal–Wallis test; *Figure 2—figure supplement 2C*). Notably, SST neurons were strongly responsive to visual stimulation (*Figure 2—figure supplement 3(iii*; mean $\Delta F/F_0$ during locomotion = 0.06 ± 0.02 darkness versus 0.25 ± 0.05 visual stimulation; p=0.001, n = 11, Wilcoxon signed rank test). These results indicate that most SST neurons respond to visual stimuli and, in addition to this visual response, they become responsive to locomotion. In darkness, however, they have low spontaneous activity and are largely non-responsive to locomotion (*Figure 2E*).

### PV neurons

Finally, PV neurons (n = 199 in 13 mice) were strongly responsive to locomotion in both sensory contexts (*Figure 2A(iv),B(iv)*, *Figure 2—figure supplement 1B(iv)*; dark: mean $\Delta F/F_0$ = 0.33 ± 0.07 locomotion versus 0.13 ± 0.02 stationary; p=0.001;, visual stimulation: mean $\Delta F/F_0$ = 0.41 ± 0.08 locomotion versus 0.16 ± 0.03 stationary; p<0.0001; n = 13 Wilcoxon signed rank test), with no significant difference between sensory conditions (*Figure 1E*, *Figure 2C(iv),D(iv)*; mean of median LMI: 0.32 ± 0.06 darkness versus 0.35 ± 0.04 visual stimulation; p=0.663, n = 13, Kruskal–Wallis test). Similarly, the cross-correlation between running speed and calcium transients showed a positive peak around time zero both in darkness and during visual stimulation, with no significant difference (p=0.778; n = 13, Kruskal–Wallis test; *Figure 2—figure supplement 2*).

## Modulation of neuronal responses by locomotion during patterned and non-patterned visual stimuli

Isoluminant grey screen stimulation is commonly used to record so called 'spontaneous activity' of neurons in the visual cortex. Since our results showed different locomotion responses in the dark and during the presentation of drifting gratings, we tested whether this difference was due to the presence of patterned visual stimuli or, more simply, to the presence of light (*Figure 2—figure supplement 1*). We quantified the amplitude of fluorescence changes during stationary and locomotion periods in all three contexts: darkness, grey screen and drifting gratings. We did not find any significant difference for any of the inhibitory populations (VIP, SST and PV neurons) between the two types of visual stimulation (gratings vs grey screen; *Figure 2—figure supplement 1C*). For excitatory neurons, we found a lower LMI during the presentation of a grey screen than during drifting grating presentation (mean of median LMI: 0.17 ± 0.02 grey versus 0.19 ± 0.02 visual stimulation; p=0.033, n = 12, Kruskal–Wallis test; *Figure 2—figure supplement 1C(i)*). Locomotion responses for each type of visual stimulus (gratings vs grey screen) were still significantly higher than during darkness (mean of median LMI: 0.07 ± 0.02 dark versus 0.17 ± 0.02 grey; p=0.007, n = 12, Kruskal–Wallis test) (*Figure 2—figure supplement 1C(i)*). These results indicate that, during visual stimulation and independently of the presence of patterned visual stimuli, excitatory, VIP, SST and PV neurons show increased activity during locomotion.

## Diversity of context-dependent locomotion responses within cell types

While comparisons of a neuronal population's LMI distribution (*Figure 2C*) indicates how, on average, that cell type is modulated by locomotion in different sensory contexts, it does not provide information about the context-dependent responses of single neurons. For instance, the average LMI could be the same in darkness and during visual stimulation even though individual neurons may have large changes in their LMI, which cancel out when considering the population as a whole.

In order to show the diversity of locomotion responses within each neuronal subtype, we examined the LMI value in darkness versus during visual stimulation for each neuron (*Figure 3A*). Neurons near the identity line show context-independent locomotion responses (similar LMI in darkness and during visual stimulation), while the other neurons changed their response to locomotion from one context to another (context-dependent responses). We first quantified this diversity by calculating the difference between the LMI value during visual stimulation and the LMI value in darkness for each neuron (*Figure 3B*). These results confirmed that VIP neurons displayed mainly context-independent locomotion responses (*Figure 3B(ii)*, narrow distribution, centered around 0), while locomotion responses of SST neurons were mainly context dependent (*Figure 3B(iii)*, broad distribution shifted towards positive values). Both excitatory and PV neuronal populations included a diversity of locomotion responses (broad distributions). To quantify the proportions of context-independent and context-dependent neurons in each cell type, we first determined the variability of the locomotion responses for each context by comparing neuronal responses across odd and even locomotion periods (*Figure 3—figure supplement 1*; see Materials and methods). We found high correlation values for all neuronal populations, both in darkness and during visual stimulation (0.676 < R < 0.944; p<0.0001), indicating a general low variability of the responses across different locomotion periods in both contexts. We determined the proportion of context-dependent neurons meeting two criteria: i) with a response that was significantly different across contexts (neurons distance from the identity line in *Figure 3A*, to estimate the error on the LMI in both dark and stimulated conditions for

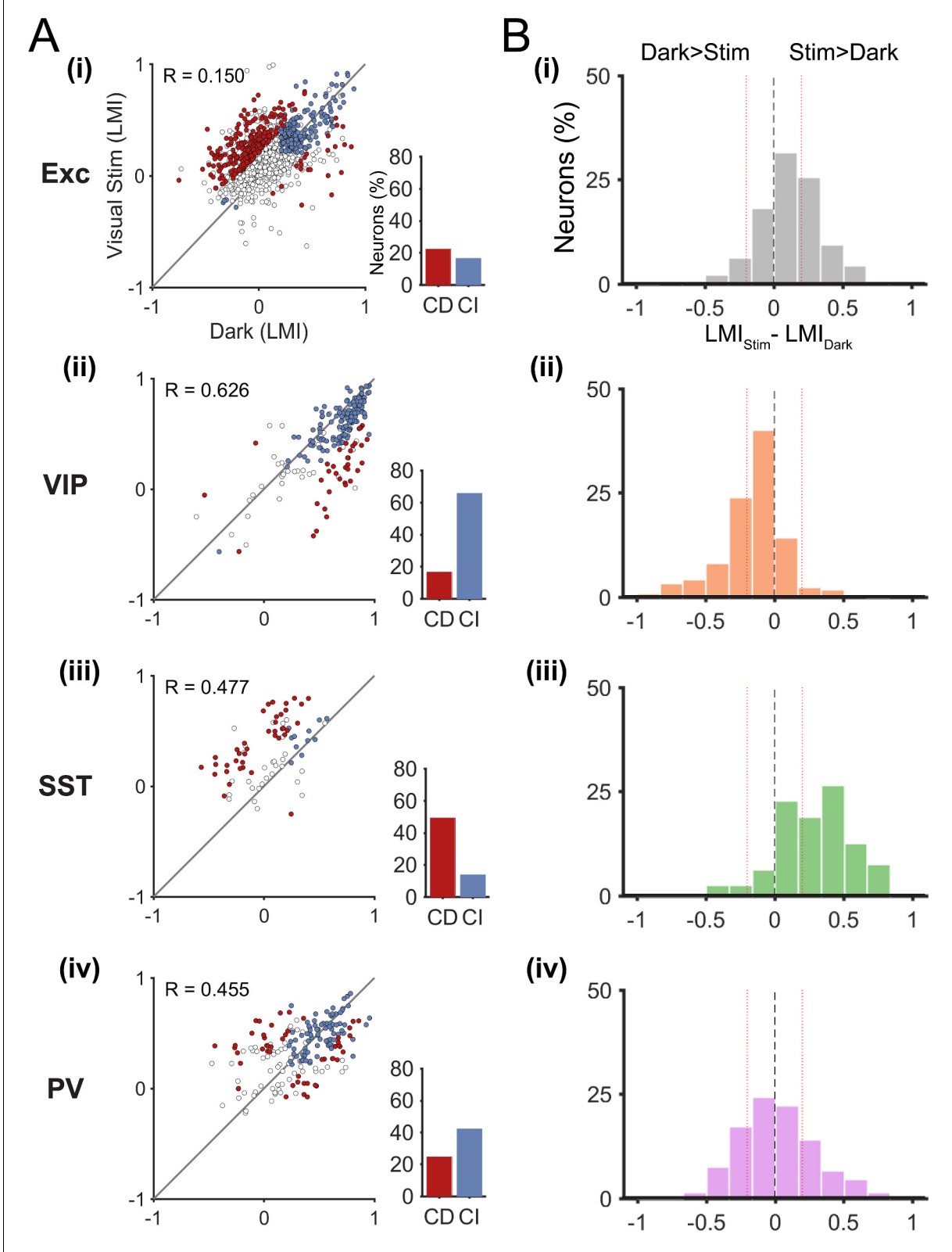

**Figure 3.** Context-dependent responses to locomotion of individual excitatory and inhibitory neurons in layer 2/3. (**A**) Left panels, scatter plots of the locomotion modulation index (LMI) of individual neurons in darkness versus during visual stimulation (gratings) with an associated Pearson correlation coefficient (R-values). Context-dependent (CD; red) and context-independent (CI; blue) locomotion responsive neurons are highlighted. Context dependency was defined for each neuron by its distance from the identity line and the variability of its locomotion responses (see Materials and
*Figure 3 continued on next page*

*Figure 3 continued*

methods and *Figure 3—figure supplement 1*). Neurons that were either non-responsive to locomotion or responded unreliably are shown as open circles. Right panels, percentages of context-dependent (CD) and context-independent (CI) neurons for each neuronal subtype. Note the high proportion of CI VIP neurons (66%), the high proportion of CD SST neurons (49%), and the diversity of both PV and excitatory (Exc) neurons. (B) Histograms of the difference between the LMI value in darkness and during visual stimulation (LMI$_{Stim}$-LMI$_{Dark}$) for each neuronal population. Negative values indicate increased responses to locomotion in darkness compared with visual stimulation, positive numbers indicate increased responses to locomotion during visual stimulation, and numbers close to 0 (within red lines; $-0.2 <$ LMI$_{Stim}$-LMI$_{Dark} < 0.2$) indicate context-independent responses.

The following figure supplements are available for figure 3:

**Figure supplement 1.** Variability of locomotion responses in darkness and during visual stimulation.
**Figure supplement 2.** Representative examples of calcium transients (ΔF/F$_0$) of context-independent and context-dependent neurons, in darkness and during visual stimulation with oriented gratings (grey bar above trace).

each neuron, bootstrapping was employed (see Materials and methods)), and ii) with low variability of locomotion responses (*Figure 3—figure supplement 1*).

These results confirm that most VIP neurons were context-independent, remaining locomotion-responsive in both sensory contexts (66%), with only 17% of neurons showing context-dependent responses (*Figure 3A(ii)*, *Figure 3—figure supplement 2*). The proportion of context-dependent neurons was the highest among SST neurons, with 49% of neurons showing context-dependent responses to locomotion (*Figure 3A(iii)*, *Figure 3—figure supplement 2*). Both excitatory and PV neurons had approximately the same proportion of context-dependent neurons (22% for excitatory and 25% for PV neurons) (*Figure 3A(i), (iv)*).

Finally, we tested whether context-dependent neurons differ from context-independent ones with regard to the following characteristics: percentage of visually responsive neurons, orientation selectivity and direction selectivity. We did not find any significant difference in any neuronal population (comparisons between context-dependent and context independent neurons for each cell type, OSI, p>0.261; DSI p>0.093, Kruskal–Wallis test), suggesting that the mechanisms underlying the modulation of locomotion responses differ from those determining the selectivity of visual responses.

## Layer 4 excitatory and inhibitory responses to locomotion are similar to layer 2/3

Layer 2/3 neurons receive sensory information from excitatory neurons in layer 4, the main thalamo-recipient layer, as well as top-down information from higher cortical areas (*Niell, 2015*). In addition, these neurons receive subcortical inputs from the dorsal lateral geniculate nucleus as well as neuro-modulatory inputs (*Polack et al., 2013*; *Fu et al., 2014*; *Lee et al., 2014*). Context-dependent loco-motion responses of layer 2/3 neurons may thus come from one of these distinct inputs or from a combination of them. By using the same approach as for layer 2/3 neurons, we recorded locomotion responses in layer 4 neurons (excitatory n = 331; VIP n = 57; SST n = 74; PV n = 109; in 6, 4, 6 and 6 mice, respectively). As in layer 2/3, we used local injections of AAVs into V1 for the Cre-inducible expression of the genetically-encoded calcium indicator GCaMP6f. However, we observed that on average the GCaMP6f labelling in layer 4 was sparser than in layer 2/3 (*Figure 1D*). Thus, we cannot exclude that we preferentially labelled subtypes of layer 4 neurons in which transduction efficiency with these AAV vectors would be higher. The quantification of locomotion responses showed no significant difference between layer 2/3 and layer 4 neurons, in any cell type, both in darkness and during visual stimulation (*Figure 4B,C*). The results showed a higher mean LMI value for PV neurons in layer 4 (0.45 ± 0.04) compared to layer 2/3 (0.35 ± 0.04) during visual stimulation. However, this did not reach significance; p=0.058, Mann-Whitney U-test). In addition, the results showed similar proportions of context-dependent responses in layer 4 as described in layer 2/3 (*Figure 4A*, see also *Figure 3A*; context-dependent neurons: Exc, L2/3: 22%, L4: 17%; VIP, L2/3: 17%, L4: 26%; SST, L2/3: 49%, L4: 42%; PV, L2/3: 25%, L4: 23%).

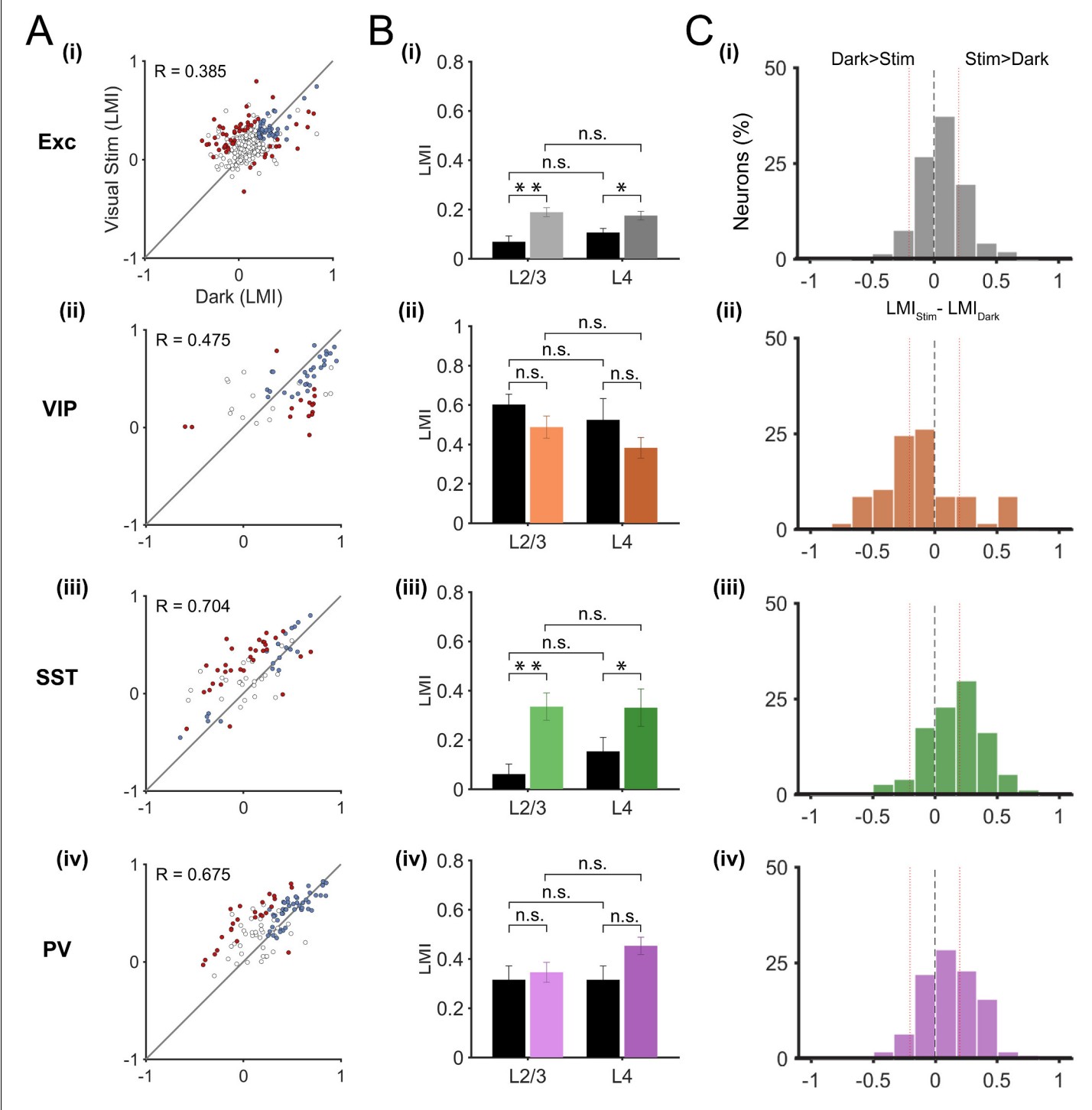

**Figure 4.** Locomotion responses of individual inhibitory and excitatory neurons in V1 cortical layer 4. (**A**) Scatter plots of locomotion modulation index (LMI) of individual neurons in darkness versus during visual stimulation (oriented gratings), with associated Pearson correlation coefficient (R-values) for excitatory (Exc; n = 331), VIP (n = 57), SST (n = 74), and PV (n = 109) neurons. Context-dependent (red) and context-independent (blue) locomotion responsive neurons are highlighted. Context dependency per neuron was defined by its distance from the identity line and its variability to locomotion periods (see Materials and methods). Neurons that were either non-responsive to locomotion or responded unreliably are shown as open circles. (**B**) Mean of the median LMI per animal and s.e.m. for layer 2/3 (L2/3) as well as layer 4 (L4), in darkness (Dark, black bars) and during visual stimulation (Stim, coloed bars) for Exc (L2/3, n = 12; L4, n = 6), VIP (L2/3, n = 12; L4, n = 4), SST (L2/3, n = 11; L4, n = 6), and PV (L2/3, n =13; L4, n =6) mice. Within each cell type, there was no significant difference (n.s., p>0.05, Mann-Whitney U test) between the median LMI across layers in either context (darkness: Exc, p=0.151; VIP, p=0.521; SST, p=0.350; PV, p=0.966; visual stimulation: Exc, p=0.750; VIP, p=0.133; SST, p=0.961; PV, p=0.058; (**C**) Histograms of the

*Figure 4 continued on next page*

*Figure 4 continued*

difference between the LMI value in darkness and during visual stimulation ($LMI_{Stim}$-$LMI_{Dark}$) for each cell type. Negative values indicate increased responses to locomotion in the dark compared with visual stimulation, positive numbers indicate increased responses to locomotion during visual stimulation, and numbers close to 0 (within red lines; $-0.2 < LMI_{Stim}-LMI_{Dark} < 0.2$) indicate context-independent responses.

## Discussion

The increased gain of visual responses during locomotion provides a model to elucidate the circuit mechanisms underlying behavioral-state dependent changes of sensory responses. In this study, we found that the modulation of neuronal activity by locomotion is context-dependent and cell type specific, in layer 2/3 and layer 4 of mouse V1. During periods of visual stimulation, locomotion increases the activity of excitatory neurons as well as of three classes of inhibitory neurons (VIP, SST, PV; *Figure 2E*). These results indicate that the enhancement of excitatory neuron visual responses during locomotion does not result from the inhibition of SST neurons, in mouse V1. Our findings thus challenge the generality of a disinhibitory circuit involving VIP, SST and pyramidal neurons for the gain control of sensory responses by behavioral state.

### Relationship between somatic fluorescence changes and spiking activity in different neuronal types and behavioral contexts

In this study, we used the relative changes in fluorescence of the genetically-encoded calcium indicator GCaMP6f as a reporter of the spiking activity of cortical neurons (*Chen et al., 2013*). For a given fluorescent calcium indicator, the relationship between the amplitude of somatic fluorescence changes and the number of spikes can be affected by a number of factors including the concentration of calcium buffers in the soma, the balance between calcium influx and efflux as well as calcium release from internal stores (*Grienberger et al., 2012*). Consequently, potential confounding factors in the present study would be (1) different intracellular calcium dynamics in different types of inhibitory neurons as well as (2) a higher increase of cytosolic free calcium concentration for the same number of spikes during locomotion compared to stationary periods. Considering that neuromodulators can regulate calcium influx (*Fucile, 2004*; *Shen and Yakel, 2009*), this second possibility may result from the action of neuromodulators released during locomotion that would increase the amount of calcium entering the neuron in response to each spike. In that case, for the same number of spikes, the increase in fluorescence of our calcium indicator would be higher during locomotion than during stationary periods.

Without an independent readout of the spiking activity for each neuronal type in each behavioral context, we cannot exclude that the relationship between fluorescence transients and the number of spikes differ between different neurons and different contexts. However, the comparison of our results (mean $\Delta F/F_0$, *Figure 2—figure supplement 1B*, 'stim' column) with spiking frequencies published in a previous study (see Supplementary Table 3 of *Polack et al., 2013*) in mouse V1 strongly suggests that somatic GCaMP6f fluorescence changes do reflect changes in spiking activity related to locomotion. For the same neuronal populations (layer 2/3 Excitatory, SST and PV neurons; layer 4 Excitatory neurons) and visual stimulation condition (drifting gratings), both data sets show the same relative change in signal during locomotion compared to stationary periods (corresponding to an approximate doubling of activity during locomotion for all three cell types). This similarity suggests that somatic GCaMP6 fluorescence changes during locomotion do reflect changes in spiking activity, at least in these cell types during visual stimulation.

### Comparison with previous findings: locomotion responses differ in darkness and during visual stimulation

In this study, we found that SST activity increased with locomotion during visual stimulation. This is in line with previous electrophysiological recordings of SST neurons (*Polack et al., 2013*) but in contradiction with the current disinhibitory model that relies on the inhibition of SST neurons during locomotion (*Figure 1B*; *Fu et al., 2014*). Our results provide an explanation for these discrepancies since the aforementioned electrophysiological recordings were acquired during visual stimulation whereas imaging of SST activity was done in the dark (*Fu et al., 2014*). The disinhibitory model was

based on the assumption that the locomotion-driven response of SST neurons would be similar in the dark and during visual stimulation (*Fu et al., 2014*). The same assumption was made in the interpretation of membrane potential fluctuations of VIP and SST neurons recorded during the presentation of a blank screen (*Reimer et al., 2014*). While VIP neurons were reliably depolarized during running, the SST population was heterogeneous. The authors distinguished two populations of SST interneurons (see Supplementary Figure 5C of *Reimer et al., 2014*): Type I cells were inhibited by running while Type II cells were depolarized. Importantly, spiking activity of SST neurons was not reported and it is thus not clear how the membrane potential fluctuations relate to spiking activity.

Our findings regarding the locomotion responses of SST neurons in darkness are consistent with the previous imaging study performed in similar conditions (*Figure 2—figure supplement 2* of the present study compared to *Figure 3* and Figure S3 of *Fu et al., 2014*) as well as with the heterogeneity of membrane potential fluctuations of SST neurons during locomotion (*Reimer et al., 2014*). We cannot exclude the possibility that a disinhibitory circuit may underlie the activity of a small fraction of neurons in darkness: the majority of VIP neurons increase their activity with locomotion, while a small proportion of SST neurons are inhibited during locomotion, potentially leading to the increase in activity of some pyramidal neurons. However, the results obtained in darkness show that the majority of SST neurons are not responsive to locomotion at all, challenging the generality of a disinhibitory circuit acting through the inhibition of SST neurons. With visual stimulation, the inconsistency of the disinhibitory model is even stronger since the vast majority of SST neurons increase their activity with locomotion (see *Figure 2B(iii)*). Consequently, the results obtained during visual stimulation (present study and *Polack et al., 2013*) are incompatible with a model in which VIP neurons disinhibit excitatory neurons by inhibiting SST neurons. Additionally, while the vast majority of VIP neurons are context-independent with regard to their locomotion response, excitatory neurons show significantly increased locomotion responses during visual stimulation compared to darkness conditions. Therefore, the context-dependent responses of excitatory neurons do not result from a disinhibitory circuit initiated by VIP neurons.

An appealing aspect of the disinhibitory model was the idea of a canonical circuit for gain modulation of sensory responses (*Pi et al., 2013*). While the connectivity may be canonical, we show that the circuit activity can strongly differ depending on the behavioral context. Therefore, functional properties of inhibitory neurons should not be generalized from one context to the next, and caution should be taken when inferring connectivity from functional recordings obtained in a specific behavioral context.

## Alternative circuit mechanisms for behavioral-state modulation of visual responses in V1

Our results indicate that, in addition to the activation of VIP neurons during locomotion, other pathways are involved in linking locomotion and visual responses in V1. We suggest that neuromodulatory inputs triggered by locomotion would not only activate VIP neurons through nicotinic acetylcholine receptors as previously shown (*Alitto and Dan, 2012*; *Arroyo et al., 2014*; *Fu et al., 2014*), but would also directly activate PV, SST, and excitatory neurons. Previous work has demonstrated cholinergic facilitation of cortical inhibitory neurons (*Kawaguchi, 1997*; *Xiang et al., 1998*; *Arroyo et al., 2012*; *Alitto and Dan, 2012*), including SST neurons (*Kawaguchi, 1997*; *Fanselow et al., 2008*; *Xu et al., 2013*; *Chen et al., 2015*). Similarly, in vitro studies have shown that norepinephrine can depolarize both excitatory (*McCormick et al., 1993*; *Kirkwood et al., 1999*) and inhibitory (*Kawaguchi and Shindou, 1998*) cortical neurons. Finally, in vivo studies have shown that neuromodulatory inputs, cholinergic and noradrenergic, can control the gain and signal-to-noise ratio of V1 excitatory neurons during locomotion (*Pinto et al., 2013*; *Polack et al., 2013*; *Bennett et al., 2014*; *Lee et al., 2014*). We suggest that in darkness, the effect of neuromodulatory inputs remains subthreshold in SST neurons. During visual stimulation, SST neurons are strongly activated and the effect of neuromodulatory inputs becomes suprathreshold. In agreement with the known intra-cortical connectivity in mouse V1 (*Figure 1A*; *Pfeffer et al., 2013*; *Jiang et al., 2015*), our findings support this neuromodulatory hypothesis. In darkness, VIP and PV neurons are activated by locomotion and inhibit SST and excitatory neurons, preventing their activation by locomotion-dependent inputs. During visual stimulation, SST and excitatory neurons are activated: they overcome the intra-cortical inhibition by VIP and PV neurons and become responsive to direct locomotion-dependent inputs. Since SST neurons provide the main intra-cortical input to VIP neurons

(*Pfeffer et al., 2013*) and are strongly visually-responsive, they likely inhibit VIP neurons (or a sub-population of VIP neurons) during visual stimulation. This is consistent with the decrease in activity of a portion of VIP neurons that was observed during visual stimulation (*Fu et al., 2014*; see also *Figure 2—figure supplement 3B(ii)*).

An alternative or complementary hypothesis to the neuromodulatory pathway is that the modulation of visual inputs by locomotion already takes place in subcortical nuclei, such that the thalamo-cortical inputs received by excitatory neurons, and potentially SST neurons, would convey the increased gain of visual responses during locomotion. Indeed, recent studies have shown that projections from the dorsal lateral geniculate nucleus (*Erisken et al., 2014*; *Roth et al., 2016*) and from the thalamic latero-posterior nucleus (*Roth et al., 2016*) to V1 both convey locomotion signals.

The diversity of context-dependent responses to locomotion within SST, PV and, to a lesser extent, VIP populations indicates that there are functional sub-types within each of these interneuron populations. Based on a comprehensive analysis of morphological and electrophysiological properties of inhibitory neurons, a recent in vitro study has identified seven distinct types of cortical interneurons in layer 2/3 (*Jiang et al., 2015*). Further, in vivo characterization of the activity of these subtypes will be necessary to identify how these populations relate to the different context-dependent responses identified in the present study.

## Materials and methods

### Animals

Three Cre-driver transgenic mice lines were used to label inhibitory interneurons: *Sst*<tm2.1(cre)Zjh> (SST-Cre) [RRID:IMSR_JAX:013044], *Pvalb*<tm1(cre)Arbr> (PV-Cre) [RRID:IMSR_JAX:008069], *Vip*<tm1(cre)Zjh> (VIP-Cre) [RRID:IMSR_JAX:010908], all originally obtained from Jackson Laboratory, ME, USA. These lines were cross-bred with Rosa-CAG-LSL-tdTomato [RRID:IMSR_JAX:007914] mice. C57Bl/6 wild type mice (Jackson Laboratory, ME) were used for virus injections targeting the expression of GCaMP6 in CaMKII-expressing neurons. Mice were group housed (typically 2–4 mice) and both male and female mice were used for the experiments. All procedures were approved by the University of Edinburgh animal welfare committee, and were performed under a UK Home Office project license.

### Surgical procedures

#### Virus injections

For virus injections, 8- to 10-week-old mice were anesthetized with isoflurane (4% for induction and 1–2% maintenance during surgery) and mounted on a stereotaxic frame (David Kopf Instruments, CA). Eye cream was applied to protect the eyes (Bepanthen, Bayer, Germany) and analgesics were injected subcutaneously (Vetergesic, buprenorphine, 0.1 mg/kg of body weight). After an incision was made in the scalp, the bone surface was cleaned and a small craniotomy was performed over the left V1 (3.5 mm lateral and 1 mm anterior to lambda with an injection pipette inserted 70° from vertical and 30° from midline). Adeno-associated viruses (AAVs) were injected using a pipette with 20 µm tip diameter (Nanoject, Drummond Scientific, PA) at a speed of 10 nl min$^{-1}$ at three different depths (around 250, 400, and 600 µm deep; 50 nl per site). AAVs used in this study include: AAV1.Syn.Flex.GCaMP6f.WPRE.SV40 to label SST, PV, and VIP cells in Cre-driver transgenic mice as well as AAV1.Syn.GCaMP6f.WPRE.SV40 in tdTomato crosses (see above) and AAV1.CaMKII0.4.Cre.SV40 with AAV1.Syn.Flex.GCaMP6f.WPRE.SV40 in C57Bl/6 wild type mice (all AAVs acquired from the University of Pennsylvania Vector Core, PA). After each injection, pipettes were left in situ for an additional 5 min to prevent backflow. The skin was then sutured and mice were monitored until they recovered from anesthesia. The animals were returned to their home cage for 2–3 weeks.

#### Head-plate and imaging window

Mice were anesthetized with isoflurane (4% for induction and 1–2% maintenance during surgery) and mounted in a stereotaxic frame. Eye cream was applied to protect the eyes (Bepanthen, Bayer, Germany), analgesics and anti-inflammatory drugs were injected subcutaneously (Vetergesic, buprenorphine, 0.1 mg/kg of body weight, carprofen, 0.15 mg, and dexamethasone, 2 µg). A section of scalp was removed and the underlying bone was cleaned before a craniotomy (around 2 × 2 mm) was

made over the left V1 (centered around 2.5 mm lateral and 0.5 mm anterior to lambda). The craniotomy was then sealed with a glass cover slip and fixed with cyano-acrylic glue. A custom-built head-post was implanted on the exposed skull with glue and cemented with dental acrylic (Paladur, Heraeus Kulzer, Germany).

## Two-photon calcium imaging

Imaging was performed using a custom-built resonant scanning two-photon microscope with a Ti: Sapphire pulsing laser (Chameleon Vision-S, Coherent, CA, USA; < 70 fs pulse width, 80 MHz repetition rate) tuned to 920 nm. Using a 40X objective (0.8 NA, Nikon), 600×600 pixel images with a field-of-view of 250 × 250 µm were acquired at 40 Hz with custom-programmed LabVIEW based software (version 8.2; National Instruments, UK).

We used two-photon calcium imaging in head-fixed mice that ran freely on a cylindrical treadmill (*Figure 1C*; *Dombeck et al., 2007*). Habituation and imaging started 2–3 weeks after AAV injection. Mice were habituated to head-fixation in the dark for 45 min and began to run freely on a polystyrene cylinder (20 cm diameter, on a ball-bearing mounted axis). The mice's running speed on the circular treadmill was continuously monitored using an optical encoder (E7P, 250cpr, Pewatron, Switzerland) connected to a data acquisition device (National Instrument, UK) with custom-written software in LabView (National Instrument, UK) and analyzed in MATLAB (Mathworks, MA). Mice could run freely and spent on average 26 ± 2% of the time running in the dark and 41 ± 2% during visual stimulation (n = 48 mice, 51 sessions).

Two-photon imaging was performed at 2–3 focal planes per mouse, at cortical depths between 130 and 350 µm for L2/3 neurons and 350–500 µm for L4 neurons (cortical layers were confirmed on histological sections, see below). Laser power at the brain surface was kept below 50 mW. Mice with excessive brain movement artifacts were excluded. At each focal plane (n = 100 fields of view), 8–12 trials (60 s duration) were acquired in total darkness and 12–20 trials acquired during visual stimulation, with dark and visual stimulation trials randomly interleaved.

Visual stimuli were generated using the Psychophysics Toolbox package (*Brainard, 1997*) for MATLAB (Mathworks, MA) and displayed on an LCD monitor (51 × 29 cm, Dell, UK) placed 20 cm from the right eye, covering 104° × 72° of the visual field. Visual stimulation trials consisted of stationary full-field square-wave gratings for 4–5 s and the corresponding drifting phase for 2 s (0.03 cpd, 1 Hz, 8 equally spaced directions in randomized order, contrast 80%, mean luminance 37 cd/m$^2$). Each trial started and ended with a grey screen (isoluminance). Additional grey screen data were obtained during the presentation of an isoluminant grey screen for 5–15 s preceding the presentation of each oriented grating for 5 s (0.03 cpd, 1 Hz, 4 equally spaced orientations in randomized order, contrast 80%, mean luminance 37 cd/m$^2$).

At the end of the imaging session, red retrograde beads (Lumafluor, USA) were injected either at the surface or at 2 different focal planes at which neurons had been imaged. This red labelling was used as a structural landmark in histological sections to confirm which cortical layers had been imaged.

## Histology

Animals were transcardially perfused with 0.9% saline and 4% PFA in phosphate buffer (0.1 M). Brains were sliced with a vibratome (50 µm thick) and rinsed in phosphate buffered saline (PBS). The slices were then mounted and counterstained with either DAPI (Vectashield mounting medium, Vector Labs, UK) or NeuroTrace 640/660 fluorescent Nissl stain (1:2000; RRID:nlx_152414, Life Technologies, NY) and coverslipped. Sections were imaged with a confocal microscope (Nikon A1R, Nikon Instruments, UK) to define the boundaries of cortical layers and localize the retrograde beads injected at the imaged focal planes in vivo.

## Data analysis
### Image analysis

To correct for brain motion after image acquisition, we used 2D plane translation-based image alignment (SIMA 1.2.0, sequential image analysis; *Kaifosh et al., 2014*). Regions of interest (ROIs) corresponding to neuronal cell bodies were selected manually by inspecting down-sampled frames (2 Hz), as well as the maximum intensity projection of each imaging stack (60 s trial). The pixel intensity

within each ROI was averaged to create a raw fluorescence time series F(t). Baseline fluorescence $F_0$ was computed for each neuron by taking the fifth percentile of the smoothed F(t) (1 Hz lowpass, zero-phase, 60th-order FIR filter) over each trial ($F_0(t)$), averaged across all trials. As a consequence, the same baseline $F_0$ was used for computing the changes in fluorescence in darkness and during visual stimulation. The change in fluorescence relative to baseline, $\Delta F/F_0$ was computed by taking the difference between F and $F_0(t)$ and dividing by $F_0$. In order to remove neuropil contamination, we used nonnegative matrix factorization (NMF), which is a low rank matrix decomposition method used for demixing spatially overlapping signal sources (*Kim and Park, 2007*; *Langville et al., 2014*), as implemented in NIMFA 1.2.1 (*Žitnik and Zupan, 2012*). The Python toolboxes were run with Win-Python **2.7**.10.3. All further analyses were performed using custom-written scripts in MATLAB (Math-Works, MA).

## Analysis of locomotion responses

Changes in the position of the cylindrical treadmill (sampled at 12,000 Hz) were interpolated onto a downsampled rate of 40 Hz, matching the sampling rate of the two-photon imaging. To define stationary and locomotion periods we used the following criteria. Stationary corresponded to periods where the instantaneous speed (as measured at the 40 Hz sampling rate) was less than 0.1 cm/s. Locomotion corresponded to periods meeting three criteria: instantaneous speed $\geq$ 0.1 cm/s, 0.25 Hz lowpass filtered speed $\geq$ 0.1 cm/s, and an average speed $\geq$ 0.1 cm/s over a 2 s window centered at this point in time. Any inter-locomotion interval shorter than 500 ms was also labelled as locomotion. Stationary periods less than 3 s after or 0.2 s before a period of locomotion were removed from the analysis. The locomotion modulation index (LMI) was defined as the difference between the mean $\Delta F/F_0$ during locomotion ($R_L$) and stationary ($R_s$) periods, normalized by the sum of both values: LMI = $(R_L - R_s)/(R_L + R_s)$.

To estimate the error on the LMI in both dark and stimulated conditions for each neuron, bootstrapping with sample replacement was employed. We binned the signal into 1 s bins, each of which had only one visual stimulus and one behavioral activity (locomotion or stationary) throughout its duration. For each 1 s bin, we took the mean $\Delta F/F_0$ and regarded this value as a single sample. For periods of time which had a single stimulus and behavioral activity persisted for longer than 1 s, additional samples were drawn with intervals of no less than 2 s. This interval duration was selected based on the autocorrelation of the calcium fluorescence signal, which took approximately 2 s to fall to 0.5. The average correlation between consecutive samples of the same stimulus and activity condition was computed as a weighted average over all conditions, and was found to be R = 0.35. We then randomly selected samples of $\Delta F/F_0$ with replacement from our original set of samples. The number of samples selected in each bootstrap resample (65% = 1-R) was reduced from the total number of samples available to reflect the fact that our samples were not completely independent. This process was repeated 10000 times to obtain 95% confidence intervals for significance tests for each neuron individually. A neuron was considered significantly locomotion responsive if its 95% confidence interval was significantly different from an LMI of 0 and its value exceeded an LMI of 0.2 (at least 50% change in $\Delta F/F_0$ between locomotion and stationary).

To evaluate the variability of locomotion responses in a given context (dark or visual stimulation) for each neuron, we divided the data in two halves: we calculated separate LMI values for all odd and for all even locomotion periods (*Figure 3—figure supplement 1*). Neurons with the highest variability of locomotion responses were identified based on the difference between odd and even LMI values for each neuronal population. The 5% most variable neurons (i.e. neurons that fall outside the red dashed lines *Figure 3—figure supplement 1* for either dark or visual stimulation) were excluded from being defined as context-dependent.

## Statistics

The error bars in all graphs indicate standard error of the mean (s.e.m.) and statistics were performed with two-tailed tests. Unless otherwise stated, for statistical tests comparing the average $\Delta F/F_0$ of neurons between two contexts or behavioral states (in darkness versus during visual stimulation, or stationary versus locomotion periods) we used Wilcoxon signed-rank tests. For statistical tests comparing the distribution of LMIs and cross-correlations between visual stimulation contexts

we used the Kruskal–Wallis test (one-way ANOVA on ranks). For statistical tests comparing $\Delta F/F_0$ values across different layers, Mann-Whitney U tests were used.

For statistical tests we used the number of animals as our sample size because neuronal responses from the same mouse may be correlated and not represent independent samples. Therefore, comparing measures across neurons, rather than across animals, would incorrectly inflate the degrees of freedom with the risk of false positive results for detecting significant differences (*Galbraith et al., 2010*). This is especially relevant for 2-photon imaging studies where data from a large number of neurons are collected from a small number of animals.

## Acknowledgements

We thank Ian Duguid and his research group for advice and support on recordings in awake mice. We thank Matt Nolan and Ian Duguid for comments on earlier versions of the manuscript. We thank the GENIE Program and the Janelia Research Campus, specifically V Jayaraman, R Kerr, D Kim, L Looger, and K Svoboda, for making GCaMP6 available.

## Additional information

### Funding

| Funder | Grant reference number | Author |
| --- | --- | --- |
| European Commission | Marie Curie Actions (FP7), IEF 624461 | Janelle MP Pakan |
| University Of Edinburgh | Doctoral Training Centre in Neuroinformatics | Scott C Lowe |
| University Of Edinburgh | Graduate School of Life Sciences | Evelyn Dylda |
| Engineering and Physical Sciences Research Council | Doctoral Training Centre in Neuroinformatics, EP/F500385/1 | Sander W Keemink |
| European Commission | EuroSpin Erasmus Mundus Program | Sander W Keemink |
| Engineering and Physical Sciences Research Council | Doctoral Training Centre in Neuroinformatics, BB/F529254/1 | Sander W Keemink |
| European Commission | Marie Curie Actions (FP7), MC-CIG 631770 | Nathalie L Rochefort |
| Patrick Wild Centre | | Nathalie L Rochefort |
| The Shirley Foundation | | Nathalie L Rochefort |
| RS MacDonald Charitable Trust | Seedcorn Grant 21 | Nathalie L Rochefort |
| Wellcome Trust | Sir Henry Dale fellowship, 102857/Z/13/Z | Nathalie L Rochefort |
| University Of Edinburgh | Chancellor's fellow starting grant | Nathalie L Rochefort |
| Wellcome Trust | Sir Henry Dale fellowship, 102857/Z/13/Z | Nathalie L Rochefort |
| Royal Society | Sir Henry Dale fellowship, 102857/Z/13/Z | Nathalie L Rochefort |

The funders had no role in study design, data collection and interpretation, or the decision to submit the work for publication.

### Author contributions

JMPP, Conception and design, Acquired the data, Analysed and interpreted the data, Revised the manuscript; SCL, Developed the NMF-based neuropil correction method and analysed the data; ED,

Acquired and analysed the data; SWK, Developed the NMF-based neuropil correction method; SPC, Acquired data; CAC, Analyzed the data; NLR, Designed the experiments, Acquired the data, Analysed and interpreted the data, Wrote the manuscript with input from all authors

## Author ORCIDs
Janelle MP Pakan, http://orcid.org/0000-0001-9384-8067
Evelyn Dylda, http://orcid.org/0000-0002-1883-4498
Nathalie L Rochefort, http://orcid.org/0000-0002-3498-6221

## Ethics
Animal experimentation: All procedures were approved by the University of Edinburgh animal welfare committee, and were performed under a UK Home Office Project License (PPL No. 60/4570).

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
