## [Decision Letter]

Thank you for submitting your article "Behavioural state modulation of inhibitory activity is context-dependent and cell-type specific in mouse V1" for consideration by *eLife*. Your article has been reviewed by three peer reviewers, and the evaluation has been overseen by a Reviewing Editor and a Senior Editor.

The reviewers have discussed the reviews with one another and the Reviewing Editor has drafted this decision to help you prepare a revised submission.

Summary:

The study compares the modulation of neuronal activity in mouse visual cortex by locomotion between two conditions (contexts): in darkness and during visual stimulation. To monitor activity the authors use calcium indicators conditionally expressed in distinct neuron types in layer 2/3 and 4. During visual stimulation all cell types (VIP, SST, PV and Excitatory neurons) showed increased activity during locomotion. In darkness the response to locomotion was essentially restricted to VIP and PV cells. These interesting results challenge the generality of some recent publication and are discussed in the context of state dependent neuromodulation. The mechanisms by which visual processing is affected by locomotion has captured the curiosity of many labs. Clearly, determining how distinct types of cortical inhibitory neurons are modulated by locomotion is an important step towards this goal. The observations challenge the notion that all running-based modulations can be accounted for by a disinhibitory circuit involving VIP – SOM – pyramidal cells.

Essential revisions:

All reviewers felt the authors could get much more out of the data with more analysis. For example, the variability analysis in Figure 3 and Figure 4 is very hard to interpret without a quantification of measurement variability, i.e. how variable is the LMI between two sessions of dark and two session of visual stimulation. To estimate this, the authors could split the data in two and look at the variability between the first and second half (e.g. calculate an LMI on the first half the dark data and compare that to LMI from the second half of the dark data). Without this it is very hard to interpret what the variability means. The argument can be applied to data in Figure 3, and corresponding parts of Figure 4.

To provide stronger visual support for the hypothesis that locomotion effects differ between darkness and visual stimulation conditions, can the authors please show a scatter plot of the LMI during visual stimulation vs. darkness for the neurons?

Therefore, for Figure 3 and Figure 4, the reviewers strongly advise to choose a different format of presentation. For example, the reader might expect scatter plots where each dot is the LMI of a neuron measured during darkness and during visual stimulation. Adding mean differences across neurons and sem for the relevant comparisons could help convey the results of the statistical analyses. To address these criticisms, the authors should choose the most appropriate format and statistics for data presented in Figure 3.

Other reports have shown that the locomotion-based modulation of an (indiscriminative) population of V1 neurons is stronger during visual stimulation than during gray screen conditions (e.g. Erisken et al., 2014). Does the presence of light that make a difference or is it rather the visual stimulation with gratings? New experiments should be conducted using a gray screen (without patterned visual stimulation) to address this question.

Also, more analyses should be provided to address the following questions: Is there any characteristic feature of a cell that is locomotion responsive during stim only? Are the visual responses different of these cells than of cells that are locomotion responsive during darkness only? There is quite a bit of straightforward analysis that could be included that would significantly increase the value of the study.

By using the amplitude of calcium transients as a proxy for neuronal firing rates and comparing these transients across context and behavioral state, the authors make the assumption that both context and behavioral state will not impact the amount of calcium entering the cell in response to each spike. However if context and/or behavioral state modifies the level of neuromodulators in the cortex this assumption may be incorrect. Neuromodulators (e.g. acetylcholine) can regulate calcium influx through voltage gated calcium channels. This issue needs to be explicitly stated and thoroughly discussed in the Discussion.

The authors have correctly noticed several contradictions between previously published stories and have identified one factor (i.e. darkness vs. visual stimulation) that might explain at least some of these contradictions. While this is an important observation, the authors should be more precise and complete in their review of the existing literature. Here are a few examples: (1) Did Adesnik et al., 2012 indeed show higher responses of SOM for running vs. stationary? To my knowledge, they only show that activity of SOM interneurons is very low during anaesthesia, and much higher during running (2) Reimer et al., 2014 observed more hyperpolarized Vm of SOM interneurons during running with a blank screen (which is possibly a gray screen condition), but again this is not treated in the manuscript. (3) Contrary to the citation in l. 226, to my knowledge neither Ayaz et al. 2013 nor Saleem et al. 2013 performed targeted electrophysiological recordings of SOM interneurons. (4) Fu et al., 2014 already reported that the magnitude of VIP activation during locomotion is stronger during darkness than visual stimulation, but this is nowhere explicitly stated. (5) Most importantly, please explain which conditions and findings are replications of and agree with Fu et al. (2014) and which do not.

Interpretation of the layer 4 data is difficult for two reasons: a) layer 4 cells seem to be inherently resistant to transfection by AAVs using the standard promoters used (hsyn, camkII, ef1a). Transgene expression is typically weaker and sparser in layer 4 than in layer 2/3 and layer 5 pyramidal cells. Given the sparse expression in layer 4, it is more than likely that the expression system preferentially targets a specific subtype of layer 4 pyramidal cell. Thus it is hard to extrapolate to layer 4 in general. Note, this is not something the authors should address experimentally, but it is a caveat that should probably be discussed. b) Layer 4 could of course receives input from all other layers, thus it becomes a chicken and egg problem as to where the locomotion modulation in layer 4 comes from. The authors' data may therefore not support the conclusion of a "prominent role of subcortical inputs in both layers for the modulation of neuronal activity by locomotion" (Results section, subsection “Layer 4 excitatory and inhibitory responses to locomotion are similar to layer 2/3”). These issues should be discussed comprehensively in the revised manuscript.

[Editors' note: further revisions were requested prior to acceptance, as described below.]

Thank you for resubmitting your work entitled "Behavioral-state modulation of inhibition is context-dependent and cell type specific in mouse visual cortex" for further consideration at *eLife*. Your revised article has been favorably evaluated by a Senior editor, a Reviewing editor, and three reviewers.

The manuscript has been much improved but there is one remaining issue that need to be addressed before acceptance. During the consultation session, the reviewers thought that the Discussion needs a little more detail about how the disinhibitory circuit model fails. Specifically, compared to the previous version, there is the impression that the data became even more complicated. Specifically, based on the LMI, you concluded that the VIP modulation by locomotion is context-independent (i.e. probably consistent with the disinhibitory scheme), while the SST modulation during visual stimulation is probably breaking the scheme. The addition of the cross-correlation analysis, however, suggests that part of the observations look more in line with the disinhibitory model: in darkness, where VIP have higher cross-correlation peaks, SST have lower ones; during visual stimulation, where VIP have lower cross-correlation peaks, SST have higher ones. A similar trend is visible for the LMI but not significant.

The reviewers thought that while your primary message is clear (that the proposed disinhibitory circuit cannot explain the data), you could try to point out and discuss a bit more in which way the circuit fails. E.g. is it maybe already a failure that neither the SST LMI nor the SST cross-correlation are negative in the data, even during darkness? Why are the results different for the LMI and the cross-correlation analysis?

The manuscript will be accepted once you extend your discussion to include these suggestions.

---

## [Author Response]

*Essential revisions:*

*All reviewers felt the authors could get much more out of the data with more analysis. For example, the variability analysis in Figure 3 and Figure 4 is very hard to interpret without a quantification of measurement variability, i.e. how variable is the LMI between two sessions of dark and two session of visual stimulation. To estimate this, the authors could split the data in two and look at the variability between the first and second half (e.g. calculate an LMI on the first half the dark data and compare that to LMI from the second half of the dark data). Without this it is very hard to interpret what the variability means. The argument can be applied to data in Figure 3, and corresponding parts of Figure 4.*

We thank the reviewers for this suggestion. We have now added a quantification of the variability of locomotion responses in each context (darkness and visual stimulation) for each neuronal type in Figure 3—figure supplement 1. As suggested, we have divided the data in two halves: we calculated separate LMI values for all odd and all even locomotion periods. We found high correlation values for all neuronal populations both in darkness and during visual stimulation (0.674<R<0.943; p<0.0001, Figure 3—figure supplement 1) indicating a general low variability of the responses across different locomotion periods in both contexts. These results are presented in Results section, subsection “Diversity of context-dependent locomotion responses within cell types” Results section and in Material and Methods section (subsection “Analysis of Locomotion Responses).

We now use this estimation of variability for the classification of context-dependent neurons in Figure 3 and Figure 4 (see response to Point 2).

*To provide stronger visual support for the hypothesis that locomotion effects differ between darkness and visual stimulation conditions, can the authors please show a scatter plot of the LMI during visual stimulation vs. darkness for the neurons?*

*Therefore, for Figure 3 and Figure 4, the reviewers strongly advise to choose a different format of presentation. For example, the reader might expect scatter plots where each dot is the LMI of a neuron measured during darkness and during visual stimulation. Adding mean differences across neurons and sem for the relevant comparisons could help convey the results of the statistical analyses. To address these criticisms, the authors should choose the most appropriate format and statistics for data presented in Figure 3 and Figure 4.*

Following this suggestion, we have now included scatter plots showing the LMI value in darkness (x-axis) and the LMI value during visual stimulation (y-axis) for each neuron in Figure 3 and Figure 4. This format of presentation does indeed provide stronger visual support to our conclusions: neurons near the identity line display context-independent locomotion responses (similar LMI in darkness and during visual stimulation), while the other neurons changed their response to locomotion from one context to another (context-dependent responses – shown in red). We now quantify the proportion of context-dependent neurons by applying the 2 following criteria:

(1) a locomotion response that was significantly different across contexts (neurons distance from the identity line in Figure 3); We estimated the error on the LMI in both darkness and stimulated conditions for each neuron, by using bootstrapping.

(2) low variability of locomotion responses both in darkness and during visual stimulation (Figure 3—figure supplement 1 and see answer to Point 1).

A full description of the method is provided in Materials and methods (subsection “Analysis of Locomotion Responses”).

Additionally, in order to convey the results of our statistical tests, we have now added to each relevant figure histograms showing the values (i.e. mean and sem) that were used for each statistical test (Figure 2, Figure 2—figure supplement 1, Figure 2—figure supplement 3, Figure 4).

*Other reports have shown that the locomotion-based modulation of an (indiscriminative) population of V1 neurons is stronger during visual stimulation than during gray screen conditions (e.g. Erisken et al., 2014). Does the presence of light that make a difference or is it rather the visual stimulation with gratings? New experiments should be conducted using a gray screen (without patterned visual stimulation) to address this question.*

In the revised version of the manuscript, we have now included new data of neurons imaged during the presentation of a grey screen. The results are presented in Figure 2—figure supplement 1. We have quantified the amplitude of fluorescence changes during stationary and locomotion periods in all three contexts: darkness, grey screen and drifting gratings. We did not find any significant difference for any of the inhibitory populations (VIP, SST and PV neurons) between the two types of visual stimulation (gratings vs grey screen; Figure 2—figure supplement 2). For excitatory neurons, we found a lower LMI during the presentation of a grey screen than during drifting grating presentation (mean of median LMI: 0.17 ± 0.02 grey versus 0.19 ± 0.02 visual stimulation; p=0.033, n=12, Kruskal–Wallis test) (Figure 2—figure supplement 2(i)).

These results show that locomotion increases the response of excitatory, VIP, SST and PV neurons during visual stimulation, independently of the presence of patterned visual stimuli. These results are presented in the Results section.

*Also, more analyses should be provided to address the following questions: Is there any characteristic feature of a cell that is locomotion responsive during stim only? Are the visual responses different of these cells than of cells that are locomotion responsive during darkness only? There is quite a bit of straightforward analysis that could be included that would significantly increase the value of the study.*

Following the suggestions described in Point 1 and Point 2, we have now refined our criteria for the definition of context-dependent neurons. We have tested whether context-dependent neurons differ from context-independent ones with regard to the following characteristics: percentage of visually responsive neurons, orientation selectivity and direction selectivity. We did not find any significant difference in any neuronal population (comparisons between context-dependent and context independent neurons for each cell type, OSI, p>0.261; DSI p>0.093, Kruskal–Wallis test), suggesting that the mechanisms underlying the modulation of locomotion responses differ from those determining the selectivity of visual responses. These results are now included in the Results section, subsection”Diversity of context-dependent locomotion responses within cell types”.

We have also tested whether visual response properties would differ between the previously defined ‘stimulus only’ and ‘dark only’ neurons. We again did not find any significant difference for the proportion of visually responsive neurons, the orientation and direction selectivity in any cell type. Taking these results into account, we have followed the suggestion of Point 10, and removed the pie chart classification of cell types. We now have only two categories: significantly context dependent and context independent neurons, defined by using bootstrapping methods and variability criteria (see Materials and Methods and responses to Points 1 and 2). We believe this simplified classification better reflects the nature of the data while still demonstrating the diversity of responses that exist within each cell type.

*By using the amplitude of calcium transients as a proxy for neuronal firing rates and comparing these transients across context and behavioral state, the authors make the assumption that both context and behavioral state will not impact the amount of calcium entering the cell in response to each spike. However if context and/or behavioral state modifies the level of neuromodulators in the cortex this assumption may be incorrect. Neuromodulators (e.g. acetylcholine) can regulate calcium influx through voltage gated calcium channels. This issue needs to be explicitly stated and thoroughly discussed in the Discussion.*

We agree and we have now included a section at the beginning of the discussion in which we clearly present and discuss this issue (subsection “Relationship between sp,atic and fluorescence changes and spiking activity in different neuronal types and behavioral contexts).

Without an independent read out of the spiking activity for each neuronal type in each behavioural context, we cannot exclude that the relation between fluorescence transients and number of spikes differ between different neurons and different contexts. However, the similarity of our results (mean df/F0, Figure 2—figure supplement 1, ‘stim’ column) with changes in spiking frequencies published in a previous study (see Supplementary Table 3 of Polack et al., 2013) in mouse V1 strongly suggests that somatic GCamp6f fluorescence changes do reflect changes in spiking activity related to locomotion. For the same neuronal populations (layer 2/3 Excitatory, SST and PV neurons and layer 4 Excitatory neurons) and visual stimulation condition (drifting gratings), both data sets show the same relative increase of activity during locomotion compared to stationary periods (corresponding to an approximate doubling of activity during locomotion for all three cell types). This similarity suggests that somatic GCamp6 fluorescence changes during locomotion do reflect changes in spiking activity, at least in these cell types during the presentation of drifting gratings.

*The authors have correctly noticed several contradictions between previously published stories and have identified one factor (i.e. darkness vs. visual stimulation) that might explain at least some of these contradictions. While this is an important observation, the authors should be more precise and complete in their review of the existing literature. Here are a few examples: (1) Did Adesnik et al., 2012 indeed show higher responses of SOM for running vs. stationary? To my knowledge, they only show that activity of SOM interneurons is very low during anaesthesia, and much higher during running (2) Reimer et al., 2014 observed more hyperpolarized Vm of SOM interneurons during running with a blank screen (which is possibly a gray screen condition), but again this is not treated in the manuscript. (3) Contrary to the citation in l. 226, to my knowledge neither Ayaz et al. 2013 nor Saleem et al. 2013 performed targeted electrophysiological recordings of SOM interneurons. (4) Fu et al., 2014 already reported that the magnitude of VIP activation during locomotion is stronger during darkness than visual stimulation, but this is nowhere explicitly stated. (5) Most importantly, please explain which conditions and findings are replications of and agree with Fu et al. (2014) and which do not.*

We have modified the text in order to be more precise in our statements:

(1) We agree that the sentence was misleading. We now cite this article in the discussion: ‘These results are consistent with electrophysiological recordings of SST neurons performed during running periods and showing an increase in the response of layer 2/3 SST neurons with stimulation of the receptive field surround, in mouse V1 (Adesnik et al. 2012)’. (Discussion section, subsection “Comparison with previous findings[…]”)

(2) Reimer et al. (2014) observed two populations of SOM interneurons in V1, that they describe as: ‘’SOM Type I and Type II cells respond differently during active behaviour (running and whisking). Type I cells are inhibited by running while Type II cells are depolarized’’). However, the authors indicate that: ‘’we excluded Type II SOM+ cells in subsequent analyses’’ (page 358; Reimer et al., 2014). Importantly, spiking activity of SOM neurons is not described in this article that only presents changes in membrane potential (Vm). It is thus not clear how the (mild) hyperpolarization of Type 1 neurons during locomotion relates to the spiking activity. These results were obtained during the presentation of a blank screen. Our results obtained in darkness are consistent with the heterogeneity of membrane potential fluctuations of SST neurons described in the Reimer et al. (2014) article.

We have included a discussion of these results in the revised version of the manuscript (Discussion section, subsection “Comparison with previous findings[…]”).

(3) We apologize for this mistake due to an error in editing the references. We have modified the sentence and removed the 2 citations.

(4) and (5) For a direct comparison of our data with the results published in the Fu et al. (2014) paper, we have now included in the revised manuscript a supplementary figure (Figure 2—figure supplement 3) in which our data are analysed in the same way as in the Fu et al. (2014) article. We calculated the cross-correlation between calcium signals and running speed for each cell type, as in Figure 2 and Figure 3 of Fu et al., (2014). In the revised version, we have now included a section in the Discussion comparing directly our data with the results presented in the Fu et al., 2014 article (Discussion section, subsection “Comparison with previous findings[…]”)

*Interpretation of the layer 4 data is difficult for two reasons: a) layer 4 cells seem to be inherently resistant to transfection by AAVs using the standard promoters used (hsyn, camkII, ef1a). Transgene expression is typically weaker and sparser in layer 4 than in layer 2/3 and layer 5 pyramidal cells. Given the sparse expression in layer 4, it is more than likely that the expression system preferentially targets a specific subtype of layer 4 pyramidal cell. Thus it is hard to extrapolate to layer 4 in general. Note, this is not something the authors should address experimentally, but it is a caveat that should probably be discussed. b) Layer 4 could of course receives input from all other layers, thus it becomes a chicken and egg problem as to where the locomotion modulation in layer 4 comes from. The authors' data may therefore not support the conclusion of a "prominent role of subcortical inputs in both layers for the modulation of neuronal activity by locomotion" (Results section, subsection “Layer 4 excitatory and inhibitory responses to locomotion are similar to layer 2/3”). These issues should be discussed comprehensively in the revised manuscript.*

We agree and we have modified the text accordingly (Results section, subsection “Layer 4 excitatory and inhibitory responses to locomotion are similar to layer 2/3”). We now state specifically that the GCamp6f labelling in layer 4 is sparser than in layer 2/3 and that we may preferentially target a subtype of layer 4 neurons. We have removed our conclusion of a *"prominent role of subcortical inputs in both layers for the modulation of neuronal activity by locomotion"* and discuss the potential input pathways that modulate the activity of both layer 2/3 ad layer 4 neurons during locomotion.

[Editors' note: further revisions were requested prior to acceptance, as described below.]

*The reviewers thought that while your primary message is clear (that the proposed disinhibitory circuit cannot explain the data), you could try to point out and discuss a bit more in which way the circuit fails. E.g. is it maybe already a failure that neither the SST LMI nor the SST cross-correlation are negative in the data, even during darkness? Why are the results different for the LMI and the cross-correlation analysis?*

We have now extended our discussion in order to emphasize how the disinhibitory circuit fails (Discussion section, subsection “Comparison with previous findings[…]”), stating more clearly how our results are incompatible with the disinhibitory circuit model.

Regarding the cross-correlation analysis, we agree that the presentation of the results was confusing. The LMI and cross-correlation analyses both show similar results regarding the relative changes of VIP and SST responses to locomotion in darkness and visual stimulation. The only difference in the previous submission was in the level of statistical significance. The cross-correlation analysis was initially included following the suggestion of the reviewers to provide a direct comparison between our data and those published previously (Fu et al. 2014). With this aim, we used the same statistical approach (pooling all neurons together) as in the Fu et al., 2014 article and the difference between dark and visual stimulation became significant for VIP neurons (whereas it is not the case in the LMI analysis in which statistics are performed across animals). We agree that this was confusing and we have now applied the same statistical approach for our cross-correlation data as for the rest of the article. As we discussed in our answer to point 20 in our first revision, we think that using the number of animals as our sample size is more suitable. Comparing measures across neurons, rather than across animals, would incorrectly inflate the degrees of freedom with the risk of false positive results for detecting significant differences. The justification of this statistical approach is clearly described in the Materials and methods section (Materials and methods section, subsection 'Statistics').

As a result, using consistent statistical criteria, there is no difference between LMI and cross-correlation analyses: they both show that during visual stimulation, VIP neurons have lower LMI and cross-correlation peaks (without reaching significance), while SST and excitatory cells have significantly higher LMI and cross-correlation peaks compared to the dark condition.

We suggest that this pattern of activity results from the fact that SST neurons provide the main intra-cortical input to VIP neurons (Pfeffer et al., 2013). Since SST neurons are strongly visually-responsive, they likely inhibit VIP neurons (or a subpopulation of VIP neurons) during visual stimulation, resulting in a decrease of activity in some VIP neurons. This is now explained in the Discussion (subsection “Alternative circuit mechanisms for behavioral-state modulation of visual responses in V1”).